# Cross-classified Multilevel Analysis of Individual Heterogeneity and Discriminatory Accuracy (MAIHDA) to evaluate hospital performance: the case of hospital differences in patient survival after acute myocardial infarction

Merida Rodriguez-Lopez [1,2] Juan Merlo [1,3] Raquel Perez-Vicente,[1] Peter Austin,[4,5,6] George Leckie[7]

For numbered affiliations see end of article.

**Correspondence to**
Dr Merida Rodriguez-Lopez;
merida.rdguez@gmail.com

## ABSTRACT

**Objective** To describe a novel strategy, Multilevel Analysis of Individual Heterogeneity and Discriminatory Accuracy (MAIHDA) to evaluate hospital performance, by analysing differences in 30-day mortality after a first-ever acute myocardial infarction (AMI) in Sweden.

**Design** Cross-classified study.

**Setting** 68 Swedish hospitals.

**Participants** 43 247 patients admitted between 2007 and 2009, with a first-ever AMI.

**Primary and secondary outcome measures** We evaluate hospital performance by analysing differences in 30-day mortality after a first-ever AMI using a cross-classified multilevel analysis. We classified the patients into 10 categories according to a risk score (RS) for 30-day mortality and created 680 strata defined by combining hospital and RS categories.

**Results** In the cross-classified multilevel analysis the overall RS adjusted hospital 30-day mortality in Sweden was 4.78% and the between-hospital variation was very small (variance partition coefficient (VPC)=0.70%, area under the curve (AUC)=0.54). The benchmark value was therefore achieved by all hospitals. However, as expected, there were large differences between the RS categories (VPC=34.13%, AUC=0.77).

**Conclusions** MAIHDA is a useful tool to evaluate hospital performance. The benefit of this novel approach to adjusting for patient RS is that it allowed one to estimate separate VPCs and AUC statistics to simultaneously evaluate the influence of RS categories and hospital differences on mortality. At the time of our analysis, all hospitals in Sweden were performing homogeneously well. That is, the benchmark target for 30-day mortality was fully achieved and there were not relevant hospital differences. Therefore, possible quality interventions should be universal and oriented to maintain the high hospital quality of care.

### Strengths and limitations of this study

► We provide a new analytical tool for analysing hospital performance based on Multilevel Analysis of Individual Heterogeneity and Discriminatory Accuracy (MAIHDA).

► Cross-classified MAIHDA disentangles the specific role of the patient-mix versus the hospital when analysing quality outcomes.

► We used a risk score to adjust for differences in patient-mix across hospitals. However, it is not a perfect instrument to quantify the true severity and mortality risk of a patient.

► MAIHDA allows analysts to identify whether target or universal interventions are most appropriate to improve the quality of care.

► We provide a three-step strategy to achieve a complete analysis of hospital performance. However, more elaborated strategies are also possible.

## INTRODUCTION
### Hospital effects

When evaluating institutional (eg, hospital) performance in healthcare, traditional studies make two implicit assumptions. First, it is assumed that over and above patient characteristics, the hospital context exerts a general, shared effect on all patients at the hospital. This general hospital-context effect is argued to reflect the influence of many factors, for instance, hospital administration, access to resources, specialised knowledge, implementation of methods for disease management and adoption of guidelines and pathways for patient treatments. Second, it is often assumed that the general

hospital-context effect can be measured by quantifying differences between hospital averages in certain quality indicators. Therefore, the focus of the analysis is based on the interpretation of tables, funnel plots, control charts, 'league tables' or similar, where hospitals are ranked according to different quality indicators such as their average 30-day mortality after acute myocardial infarction (AMI).[1] Occasionally such analyses are accompanied by an estimation of the reliability of the ranking ('*rankability*'),[2] but more often than not the focus of analysis remains on hospital averages.

## Multilevel Analysis of Individual Heterogeneity and Discriminatory Accuracy

Recently, MultilevelAnalysis of Individual Heterogeneity and Discriminatory Accuracy (MAIHDA) has been proposed as a novel strategy for evaluating hospital performance.[3] In contrast with most traditional studies, hospital MAIHDA simultaneously focusses on both hospital averages *and* patient heterogeneity around such averages. In MAIHDA, the fundamental statement is that patient and hospital variation should not be analysed separately. Rather, we need to consider that the total individual outcome variance can be partitioned into variance components operating at different levels of analysis.[4] From this perspective, hospital differences are not measured as the difference between hospital averages, but as the hospital general contextual effect (GCE). That is, the share of the total individual variance in patient outcomes that is at the hospital level. This definition aligns with that for the variance partition coefficient (VPC) in multilevel modelling. The greater the GCE, the more important the hospital context is for explaining variation in individual outcomes.[5–7] This idea is also closely related to the notion of discriminatory accuracy developed for the evaluation of the performance of prognostic and screening markers in medicine.[8 9] It is therefore possible to also use measures of discriminatory accuracy such as the area under the receiving operator characteristics curve (AUC), to quantify the hospital GCE.[10] See elsewhere for an extended explanation of the GCE concept.[3 7 11] In this article, we argue that the systematic application of measures of variance and discriminatory accuracy is of fundamental relevance for meaningful performance evaluations.[3 5 11–13]

## Cross-classified MAIHDA

Hospital comparisons are usually adjusted for 'patient-mix' using a risk score (RS). In traditional multilevel analysis of hospital performance, patient RS effects are modelled as fixed effects (eg, by entering the set of RS categories as a series of dummy variables) while the hospital effects are modelled as random effects. In contrast, in the cross-classified MAIHDA approach both the RS and the hospitals are modelled as random effects. Readers familiar with the traditional application of multilevel modelling may query the treatment of RS categories as random effects. For example, while we can think of the hospitals as a sample drawn from the set of all possible hospitals, it

proves harder to conceptualise the RS categories in this way as there is not a large population of RS categories from which they are drawn. This is, however, a philosophical, rather than a practical question. In fact, when studying hospitals in a country the hospitals are never a sample of an infinite super population of hospitals but a concrete set of facilities in a specific setting. Furthermore, many multilevel studies observe and analyse all the hospitals in a country in their data, and the total number of hospitals may not prove that large, yet here too the hospital effects will be treated as random effects. As discussed by Snijders and Bosker, when defining the random intercept model,[14] p. 45, the random effects model can be applied even when the idea of an infinite superpopulation is less evident. This approach is currently being applied when performing intersectional MAIHDA in social epidemiology.[15]

The cross-classified approach provides several advantages over the traditional hierarchical multilevel approach. First, the cross-classified MAIHDA is parsimonious as it includes only one random parameter for the $n$ RS categories rather than the $n-1$ dummy variables as in the fixed effects approach. Second, the cross-classified approach provides separate VPCs and AUCs for RS categories and for the hospital, allowing their magnitude to be contrasted. Thus, in contrast to the fixed-effects approach, it allows the importance of patient-mix versus the hospital effects to be communicated on a common metric. In addition, hospital MAIHDA provides all the usual advantages of multilevel models. For instance, by providing reliability weighted hospital averages (shrunken residuals), it reduces the concern of monitoring outcome measures based on small hospital caseloads which otherwise may lead to extreme and unstable hospital rankings and, therefore, unreliable performance evaluation.[16 17] Both hierarchical and cross-classified MAIHDA are nowadays easy to implement in available software such as MLwiN that can be run from both within Stata (runmlwin)[18] and within R (R2MLwiN).[19]

Finally, for binary patient outcomes, such as 30-day mortality after AMI, multilevel analyses can be performed using a simple contingency table or matrix with strata defined by combinations of the hospitals and the RS categories capturing patient-mix. The only information required for the analysis is the overall number of patients and the number of AMI cases in each hospital-RS stratum. This aggregated approach maintains the joint distribution of the hospitals-RS information and provides the same model results (parameter estimates, SEs, fit statistics and predictions) as when analysing the underlying individual level data. The aggregated approach also allows computationally efficient (fast) estimation as it allows analysing thousands of patients' outcomes using a data set consisting of just a few hundred strata. A further benefit of the aggregated approach is that the data can be shared since its aggregated presentation reduces ethical problems of confidentiality (statistical disclosure is not at risk). This in turn, improves the

transparency of the research and facilitates the replication of the analysis and encourages the sharing of data to compare hospital performance between different settings.

## The aim of this study

The aim of this study was to demonstrate a novel statistical approach (MAIHDA) to evaluate hospital performance and to orientate stakeholders to make assertive data informed decisions using a three-step framework. We do so by analysing differences in 30-day mortality among patients admitted to the Swedish hospital with a first-ever AMI between 2007 and 2009.

## METHODS

### Study population

This is a cross-sectional study. We used information from the Swedish Patient Register[20] and from the Cause of Death Register[21] (National Board of Health and Welfare) as well as from Population Register[22] (Statistics Sweden). To ensure the anonymity of the subjects, the Swedish authorities transformed the personal identification numbers of the individuals[23] into arbitrary personal numbers before delivering the research databases to us, and we linked the databases using the anonymised identification number.

From the research database we identified all 47 462 patients with a discharge diagnosis of AMI coded as I21 according the 10[th] edition of the International Classification of Diseases (ICD), admitted to Swedish hospitals between 2007 and 2009 and being 45 to 80 years old. The flow diagram of the study population is shown in figure 1.

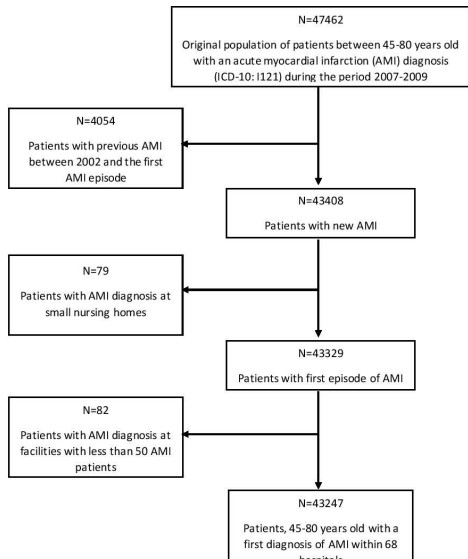

**Figure 1**  Flow diagram showing the selection of patients with first diagnosis of AMI 2007 to 2009 who were included in the study population. ICD-10,International Classification of Diseases, 10[th] Edition.

## Ethical statement

This research was done without patient involvement. The Regional Ethics Review Board in southern Sweden (# 2012/637) as well as the data safety committees from the National Board of Health and Welfare and from Statistics Sweden approved the construction of the database used in this study.

## Data accessibility

The original databases used in our study are available from the Swedish National Board of Health and Welfare and Statistics. In Sweden, register data are protected by strict rules of confidentiality[24] but can be made available for research after a special review that includes approval of the research project by both an Ethics Committee and the authorities' own data safety committees. The Swedish authorities under the Ministry of Health and Social Affairs do not provide individual level data to researchers abroad. Instead, they normally advise researchers in other countries to cooperate with Swedish colleagues and analyse data in collaboration according to standard legal provisions and procedures.

However, in the approach we propose, it is technically possible to perform the analysis using a simple table defined by hospital and categories of risk score. The aggregated information as well as the additional encryption of the hospital name fully anonymised the table, which prevents the backwards identification of individuals even when very few patients are in a single cell of the table. Therefore, to increase transparency and facilitate the replication of our analysis we provide the table as a Stata data set (online supplemental file 1) and a fully annotated Stata do-file to allow the replication of the analyses (online supplemental file 2). We also provide the table as a CSV file along with an R Script (online supplemental file 3,4).

## Assessment of variables

### Patient outcome

The study outcome was all-cause mortality within 30 days after admission to the hospital (coded yes vs no) due to AMI.

### Risk score for mortality

An inherent difficulty when investigating quality outcome indicators such as mortality is the threat of confounding due to differences in patient-mix across hospitals. The geographical areas covered by the hospital may vary in the demographical and disease characteristics of the patients. Furthermore, patients with a worse prognosis may be channelled to certain hospitals providing specialised care and this selection of patients will further confound the evaluations of hospitals differences. To reduce this form of confounding we computed a RS for 30-day mortality in the sample of AMI patients. Initially we selected a priori 40 variables including sex (man vs woman), age in years (as a quadratic function) and several previous diseases (ICD-10 codes) registered in the Swedish Patient Register.

We then modelled 30-day mortality using a single-level logistic regression in Stata 14 (Statacorp, College Station, Texas, USA), and stepwise variable selection with a significance level of 0.10 and 0.05 for removal and inclusion of variables, respectively. The variables retained in the final model were, age, arrhythmia (I48-I49), cancer (C1-D4), diseases of the cerebral arteries (I6), chronic diseases of the lower respiratory tract (J4), diabetes (E10-E14), digestive diseases (K0-K9), other types of heart disease (I3-I5), hypertension (I10-I13 I15), hearth failure (I50), injury (S00-T14), ischaemic coronary artery disease (I20-I25), lung cancer (C34), mental diseases (F0-F9) and peripheral vascular disease (I74 I80) as well as respiratory diseases (J0-J9). Finally, we obtained the predicted probability of 30-day mortality and defined it as our patient RS. We discretised the RS into 10 categories using the decile values of the RS distribution. We chose deciles to provide enough granulation of the continuous RS variable and enough categories to be included as a random effect in the multilevel model.

## Statistical analyses

We analysed 30-day mortality among 43 247 patients admitted to 68 Swedish hospitals between 2007 and 2009, with a first-ever AMI. We classified the patients into 10 RS categories for 30-day mortality and created 680 strata defined by combining hospital and RS categories. In the first step (model 1), we applied a traditional hierarchical multilevel logistic regression model with patients clustered within hospitals. In a second step (model 2), and in order to adjust for patient-mix, we performed a cross-classified multilevel model of patient outcomes with both RS categories and hospitals random effects.[25] We estimated the VPC and the AUC to evaluate differences between RS categories and between hospitals in a common way.

## Estimation methods

We performed the estimations using Markov Chain Monte Carlo (MCMC) methods, with diffuse (vague, flat or minimally informative) prior distributions for all parameters. We used quasi-likelihood methods to provide starting values for all parameters. For each model, the burn-in length was 5000 iterations. We ran the model for a further 10 000 monitoring iterations and used the resulting parameter chains from the MCMC to construct 95% credible intervals (CI) for all model predictions to communicate statistical uncertainty (online supplemental file 2). Visual assessments of the parameter chains and standard MCMC convergence diagnostics suggested that the monitored chains had converged.

## Model 1: unadjusted multilevel analysis

Model 1 was a multilevel logistic regression for patient mortality where we only include a hospital random effect to account for the variation in mortality rates across hospitals. Let, $y_i$ denote the number of deaths in hospital $j(j = 1, ..., 68)$. The model can then be written as

$$y_i \sim Binomial(n_j, \pi_j)$$
$$logit(\pi_j) \equiv log(\frac{\pi_j}{1-\pi_j}) = \beta_o + H_j$$
$$H_j \sim N(0, \sigma_H^2)$$
$$n_j \qquad \text{Formula 1}$$

The purpose of this model was to evaluate unadjusted hospital differences in average mortality risk. For this aim, we (1) ranked the hospitals according to their mortality risk; and (2) complemented this information by quantifying the size of the GCE.

### Ranking of the hospitals

To rank hospitals according to their *unadjusted* mortality rates, we predict the absolute risk ($AR_j$) of 30-day mortality and its 95% CI in each hospital. To do so, we first transformed the predicted logit of 30-day mortality into proportions ($\pi_j$) as follows

$$AR_H \equiv \pi_j \equiv logit^{-1}(\beta_o + H_j) = \frac{exp(\beta_o + H_j)}{1 + exp(\beta_o + H_j)}$$

### Measuring the hospital GCE

We estimate the hospital GCE by means of two measures

#### (i) The variance partition coefficient for the hospital level ($VPC_H$)

The $VPC_H$ can be calculated based on the latent response formulation of the model which is an approach widely adopted today in multilevel applied work.[27–31]

$$VPC_H = \frac{\sigma_H^2}{\sigma_H^2 + \frac{\pi^2}{3}}$$

Where $\frac{\pi^2}{3} \cong 3.29$ denotes the variance of a standard logistic distribution. We then multiply the $VPC_H$ by 100 and interpret it as a percentage.

The $VPC_H$, quantifies the share of the total individual differences in the latent propensity of 30-day mortality, that is, at the hospital level. The $VPC_H$ embraces the influence of the hospital context on the patient outcome without identifying any specific hospital information. However, the $VPC_H$, may also reflect differences in patient-mix between hospitals. In any case, the $VPC_H$, represents the hospital ceiling effect or potential maximum influence of the hospital attended. In the absence of confounding by patient-mix, the higher the $VPC_H$, the higher the hospital GCE is. In other words, the more relevant the hospital context is for understanding individual variation in the latent risk for 30-day mortality.

#### (ii) The area under the receiver operating characteristics curve for the hospital ($AUC_H$)

A well-known measure of discriminatory accuracy is the AUC.[10 11 32] In our case the hospital $AUC_H$ measures how well the model predicted probabilities based on the attended hospitals distinguish between two outcome categories (death within 30 days or survival). The $AUC_H$ is constructed by plotting the true positive fraction against

the false positive fraction for different thresholds of the predicted probabilities. The AUC takes a value between 1 and 0.5 where 1 is perfect discrimination and 0.5 would be equally as informative as flipping a coin (ie, the hospital information has no discriminatory accuracy).

We calculated the $AUC_H$ and to account for the different number of patients in each hospital we also calculated the weighted AUC ($AUC_{WH}$) where every patient was weighted by the inverse of the number of patients at his/her hospital. In our study both the unweighted and the weighted AUCs were almost identical so we only present the unweighted $AUC_H$.

## Model 2: patient-mix adjusted multilevel analysis

Model 2 was a cross-classified multilevel logistic regression[25] for patient mortality in which we included both hospital and RS category random effects to simultaneously account for the variation in mortality across both hospitals and RS categories and to therefore adjust the hospital effects for patient-mix. The model can be written as:

$$y_i \sim Binomial(n_j, \pi_j)$$
$$H_j \sim N(0, \sigma_H^2)$$
$$RS_k \sim N(0, \sigma_{RS}^2)$$

where $RS_K$ denotes the random effect associated with RS categories $k(k = 1, ...., 10)$, which are assumed to be normally distributed with zero mean and between-RS variance $\sigma_{RS}^2$.

The predicted logit of 30-day mortality was transformed into proportions. The mortality rates from this model are standardised and represents the rate that each hospital would have experienced if all hospitals had treated the same patients, in our case a patient with an average RS value in the population or $RS_k = 0$.

The purpose of model 2 was to evaluate patient-mix adjusted hospital differences in average mortality risk. Therefore, and analogously to model 1, we ranked the hospitals according to their RS adjusted mortality risk and complemented this information by quantifying the size of the hospital GCE net of the observed patient-mix influence. Visual inspection of the hospital and RS category predicted random effects showed the random effect normality assumptions were satisfied. (online supplemental file 2). As a measure of the patient-mix adjusted hospital GCE, we obtained the hospital $VPC_H$ as

$$VPC_H = \frac{\sigma_H^2}{\sigma_H^2 + \sigma_{RS}^2 + \frac{\pi^2}{3}}$$

We also calculated the VPC for the RS category ($VPC_{RS}$) as

$$VPC_{RS} = \frac{\sigma_{RS}^2}{\sigma_H^2 + \sigma_{RS}^2 + \frac{\pi^2}{3}}$$

The adjusted $VPC_H$ and $VPC_{RS}$ inform on the share of the total individual variance in the latent propensity of 30-day mortality, that is, at the hospital and at the RS category level, respectively, net of the influence of the other

factor. Both measures are estimated on the same scale and can therefore be directly compared with to evaluate the relative relevance of hospital versus patient-mix information when it comes to understanding patient differences in the latent propensity of death. We also calculated the adjusted AUC for the hospital level ($AUC_H$) and for RS category levels ($AUC_{RS}$) including their specific random effects when calculating the predicted probabilities. In order to compare the results from the cross-classified approach (model 2) with the traditional Hierarchical Random Intercept approach, we performed a third model including the RS categories as fixed effects (online supplemental file 2). The results from both models (2 and 3) were similar in terms of Bayesian deviance information criterion (DIC), VPC and AUC.

While patients with relatively mild conditions may have similarly good outcomes regardless of where they are treated, outcomes of the most complex patients may be affected by hospital performance. We therefore fitted a cross-classified model including a random interaction effect between the hospital and the RS ((online supplemental file 2). That is, we allowed the effect that a hospital has on their patients to vary according to the RS classification of their patients and vice versa. However, the resulting interaction classification variance was very low, suggesting that hospital attended and patient RS have additive effects on the log-odds of AMI. Consequently, we based our analysis on model 1 and 2.

## Software

All models were run in MLwiN 3.05[33] called from Stata using the *runmlwin* command.[18] We note that MLwiN can equally be called from within R using the sister R2MLwiN package[19] and so our analysis can also be replicated by readers in that statistical package.

## Evaluating hospital performance with MAIHDA

The present cross-classified MAIHDA framework extends that which was described in a previous publication aimed to the evaluation of geographical differences in health outcomes.[34] The framework proposes three steps that need be considered to achieve a complete analysis of hospital performance. However, more elaborated strategies are of course also possible, and the presented framework is open for modifications and extensions. The application of the framework in our study was as follows;

## Step 1: identifying a benchmark value and evaluating the adjusted hospital mortality rates against it

When evaluating hospital performance, we need to identify a benchmark value expressing a tolerable average level of 30-day mortality in the population of AMI patients. However, the selection of a specific benchmark is often difficult and arbitrary. We can use an *internal benchmark* defined as the $\beta_0$ obtained in model 2 (*Formula 4*). That is, the mortality rate in a hospital with an average mortality ($H_j = 0$) treating patients with an average RS ($RS_K = 0$). This choice is meaningful since comparing with a national

average seems 'fair' and being RS adjusted, tertiary care hospitals with more severe cases do not unfairly push the hospital effect towards a higher value as in the crude average rate. However, being an adjusted rate the value does not necessarily resemble the crude rates and can only be used for relative comparisons. Other adjustments are possible such as computing adjusted hospital mortality rates by holding $RS_K$ value equal to the 90th percentile of the RS random effect distribution. Another possibility would be to calculate the adjusted hospital rates that would arise if each hospital treated a nationally representative sample of patients. However, statistically, this would be more complex to do as this would require integrating out the RS random effect (via simulation) and so we do not pursue this here.

We could also use and *external benchmark* by applying a RS equation for 30-day mortality obtained in another country or in an international collaboration. This approach seems worthy for international comparisons, but it may also be inappropriate if the RS equation does not properly account for population demographical and comorbidities differences in risk, and the diagnostic criteria are not fully standardised. Finally, we could use an arbitrary rate based on previous evidence and proven expertise. Considering the data published by the Organisation for Economic Co-operation and Development (OECD)[35] and a recent review article,[36] we decide a RS adjusted 30-day mortality of less than 6% as a desirable target value for the purposes of this illustrative application. We defined three categories of achievement in relation to the benchmark, fully reached target (≤6%), closely reached target value (6% to 8%), and insufficiently reached target value (≥8%). We used caterpillar graphs to compare the RS adjusted hospital rates in relation to the benchmark value of 6%.

### Step 2: quantifying the size of the hospital differences using the VPC and the AUC

Currently there is no official guidance for assessing the magnitude of the VPC or the AUC in the context of studying hospital differences in RS adjusted hospital 30-day mortality but a practical proposal is described in table 1. This table also shows the corresponding AUC values according to the simulated relationship between

the AUC and VPC published elsewhere.[34] The proposed values are based on the authors' own experience, but further discussion is encouraged to arrive at a standard classification. Furthermore, different standards may ultimately be required and developed for different outcomes in different contexts and at different points in time. For instance, Hosmer and Lemeshow[37] suggested that AUC of 0.70 to 0.80 are 'acceptable', 0.80 to 0.90 'excellent' and 0.9 or above 'outstanding', while AUC of 0.50 suggests discrimination by chance, for example, similar to tossing a coin to decide death or alive.

### Step 3: interpreting results to evaluate performance

The two primary questions for the hospital performance evaluation were, (1) has the benchmark value been insufficiently, closely or fully reached? and (2) are there substantial differences between the hospitals, or do they perform homogeneously? To answer both questions, we created a framework (table 1) with 15 scenarios combining information on the benchmark value achievement and the size of hospital differences based on model 2.

In the best scenario (scenario A) the desired target level has been fully achieved overall (averaging across all hospitals the adjusted mortality rate is less than 6%), and hospital differences are effectively absent (the hospital GCE is effectively absent). The conclusion would be that all hospitals have performed similarly well. In the worst scenario (scenario C) the desired target level has not been achieved overall, and between-hospital differences are again absent. The conclusion would be that all hospitals have performed similarly but poorly. Observe that in both scenarios A and C, interventions only targeted to specific hospitals are not justifiable. Rather any intervention should be universal (ie, directed to all hospitals) as in both scenarios all hospitals are performing similarly. In scenario A, the intervention would be oriented to *maintaining* the overall high quality while in scenario C, the objective would be to improve the quality in *all* hospitals.

The interpretation of the scenarios in the lowest corners of the table (M and O) is driven by the very large size of the hospital differences. For example, in scenario M some hospitals may not have achieved the target level even though the average 30-day mortality is below the

**Table 1** Framework for performing hospital comparisons of 30-day mortality after acute myocardial infarction

| Size of hospital differences (hospital GCE) | | | Benchmark value achievement | | |
|---|---|---|---|---|---|
| | | | **Full** | **Close** | **Insufficient** |
| | VPC$_H$(%) | AUC$_H$ | <6% | 6% to 8% | >8% |
| Absent | 0 to 1 | 0.50 to 0.55 | A | B | C |
| Small | 1 to 5 | 0.55 to 0.61 | D | E | F |
| Moderate | 5 to 10 | 0.61 to 0.66 | G | H | I |
| Large | 10 to 20 | 0.66 to 0.72 | J | K | L |
| Very large | >20 | >0.72 | M | N | O |

AUC$_H$, hospital area under the receiving operator characteristics curve; GCE, general contextual effect; VPC$_H$, hospital variance partition coefficient.

**Table 2** Characteristics of the population of 43 247 patients with a first-ever acute myocardial infarction cared for at 68 Swedish hospitals in 2007 to 2009 and categorised in 10 groups by deciles of risk score (RS) for 30-day mortality

| | Age, mean (SD) | 67.3 (9.0) |
| --- | --- | --- |
| Female (%) | | 33 |
| Number of hospitals | | 68 |
| Number of patients at the hospital, Median (min-max) | | 636 (107–3037) |

| | Mean RS (%) | Observed mortality (%) |
| --- | --- | --- |
| Risk Score for 30 day mortality (RS) | 7.8 | 7.8 |
| Decile groups of RS | | |
| 1st | 1.7 | 0.7 |
| 2nd | 2.3 | 1.4 |
| 3rd | 2.9 | 2 |
| 4th | 3.5 | 2.5 |
| 5th | 4.2 | 3.7 |
| 6th | 5.3 | 5.9 |
| 7th | 6.9 | 8.2 |
| 8th | 9.5 | 12.2 |
| 9th | 14 | 16.1 |
| 10th | 28 | 25.5 |

benchmark value. In contrast, in scenario O, some hospitals may have achieved the target level even though the target has not been achieved overall. In these scenarios (M and O) targeted hospitals interventions are justified to focus especially on those hospitals operating above the benchmark value.

**Patient and public involvement**: No patient and public involved.

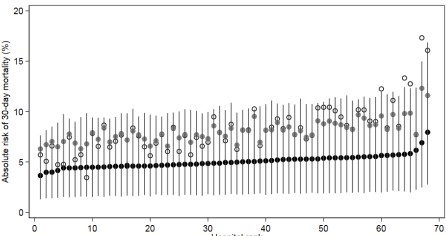

**Figure 2** Crude (white circles), reliability weighted (grey circles) and both reliability weighted and patient-mix adjusted (black circles and confidence intervals) differences in absolute risk of 30-day mortality between hospitals in the population of 43 247 patients with a first-ever acute myocardial infarction in 2007 to 2009, in Sweden.

## RESULTS

Table 2 describes the estimation sample. The mean age of the 43 247 patients with a first-ever AMI was approximately to 67 years and only one-third were women. As expected, mortality was strongly related to the RS of the patient, so that patients in the highest RS category present a mortality risk about 36 times higher than in the low RS group (25.5% vs 0.7%).

In the crude, unadjusted analysis based on simple proportions (ie, obtained without multilevel modelling), 8.18% of the patients died within a 30-day period after admission to the hospital and the crude proportions ranged from 3.45% to 17.32% across hospitals, as indicated in figure 2 (white circles). However, accounting for the reliability of the hospital information in the hierarchical multilevel analysis (Model 1), the mean unadjusted hospital rate from the multilevel model was 8.00% and ranged between 6.27% and 12.21% (grey circles). Thus, the multilevel model was able to extract the systematic variation of interest between hospitals from the observed variation which additionally includes a substantial statistical noise component due to the relatively small patient caseload per hospital (from a statistical sample size perspective). After further adjustment for patient-mix variation across hospitals in the cross-classified multilevel model (model 2) the mean hospital 30-day mortality was 4.78% and varied only between 3.44% and 7.48% (black circles) and the 95% CI (grey vertical lines) considerably overlap each other. Thus, once we performed reliability weighted estimations and adjusted for hospital differences in patient mix, very little variation in performance remained across hospitals. These adjusted mortality rates represent hospital rates as if all hospital treated the same patient with average RS (ie, $RS_k = 0$), which is a suitable approach for evaluating hospital performance.

Table 3 reports the results from the multilevel variance analysis used to estimate the GCE. In model 1, the $VPC_H$ was 1.20%, and the $AUC_H = 0.55$. Adjustment for patient-mix by including the RS categories in the cross-classified model 2 reduced the value of the $VPC_H = 0.70\%$, and slightly decreased the initial $AUC_H = 0.54$. Therefore, the hospital differences were *absent*, suggesting that hospital attended is not a driving factor in determining patient mortality. In contrast, information on the RS value of the patients was much more relevant than information on the hospital in which they were treated. The $VPC_{RS}$ was very large (ie, 34.13%) and the $AUC_{RS} = 0.77$. Figure 3 clearly depicts the differential discriminatory accuracies of the hospital and RS random effects. The hospital variance (0.03 (0.02 to 0.06)), total AUC (0.78 (0.77 to 0.79)) and Bayesian DIC (2334.2) results from model 3 were similar to model 2.

## DISCUSSION

Analysing 30-day mortality after AMI in Sweden, we illustrate the MAIHDA approach to auditing hospital performance. By considering both the size of the hospital GCE and the RS adjusted hospital 30-day mortality rates in

**Table 3** Multilevel modelling of 30-day mortality in 43 247 patients with a first-ever acute myocardial infarction in 2008 to 2009 in Sweden. Model 1 includes only a random effect for the hospital, model 2 is a cross-classified multilevel analysis including both hospital and risk score (RS) categories as random effects

| | Model 1 | Model 2 |
|---|---|---|
| Overall 30-day mortality mean (minimum–maximum), % | 8.00 (6.27–12.21) | 4.78 (3.44–7.48) |
| Variance | | |
| Hospital | 0.04 (0.02–0.07) | 0.03 (0.01–0.06) |
| RS category | | 1.96 (0.66–4.53) |
| VPC (%) | | |
| Hospital | 1.20 (0.56–2.22) | 0.70 (0.26–1.34) |
| RS category | | 34.13 (16.64–57.77) |
| AUC | | |
| Hospital | 0.55 (0.55–0.57) | 0.54 (0.53–0.55) |
| RS category | | 0.77 (0.77–0.78) |
| Total | 0.55 (0.55–0.57) | 0.78 (0.77–0.79) |
| DIC | 5480.7 | 2333.5 |

AUC, area under the receiving operator characteristics curve; DIC, Bayesian deviance information criterion; VPC, variance partition coefficient.

relation to a pre-set benchmark value, we were able to perform a more nuanced evaluation of hospital performance compared with traditional methods focussed exclusively on differences between hospital averages.

Following the framework presented in table 1, our results corresponds with the scenario A and indicate that, at the time of our analysis, all hospitals in Sweden were performing homogeneously well. That is, the overall 30-day mortality (ie, 4.78%) was under the benchmark value of ≤6% and there were no relevant hospital differences ($VPC_H$=0.70% and $AUC_H$=0.54). The ranking of hospitals in our study (figure 2) also shows that no hospitals could be statistically distinguished with any degree of certainty from the overall average mortality.

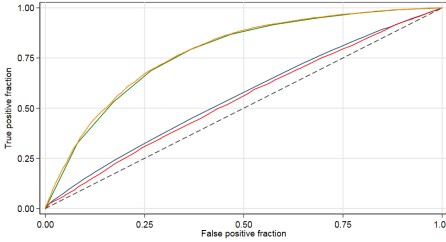

**Figure 3** Area under the receiver operating characteristics curve (AUC) constructed using the predicted probability of 30-day mortality obtained from multilevel logistic regression analysis with 47 462 patients with a discharge diagnosis of acute myocardial infarction in 2007 to 2009 in Sweden. The AUC for the unadjusted hospitals was obtained from model 1 with patient nested within hospitals (blue line). The rest of the curves are from model 2, a cross-classified multilevel model with patients nested within hospitals and categories (decile groups) of risk-score for 30-day mortality. (red line: patient mix adjusted hospitals, orange line: risk score category, green line: total AUC).

We have found similar low VPC values when investigating hospital differences in mortality after AMI admission in Ontario, Canada[38] and in Sweden[39] as well as mortality after heart failure in Sweden[12] and in Denmark.[3] The low hospital GCE suggests universal instead of targeted interventions, as all hospitals perform homogeneously. There may be, however, other patient outcomes where the hospital GCE would be much larger. For instance, when auditing adherence with guidelines for statin prescription[40 41] or process quality indicators of diabetes care like albuminuria analysis.[42]

The evaluation of institutional performance using VPC is not new.[12 39 43 44] Normand[45] states that when evaluating hospital performance if the VPC is zero *there are no hospital quality differences, that is, the chance that a patient experiences an event after being treated is the same regardless of the hospital (p. 33)*. This idea is also explicitly expressed by the committee assigned to set statistical guidelines for assessing hospital performance in USA.[46] The share of the total variance, that is, at the hospital level is crucial for evaluating performance. However, this fundamental concept needs be applied more extensively. Today, it is recognised that multilevel models (hierarchical or mixed-effect models) are the preferred methodology for provider profiling. However, the substantive analysis of components of variance still receives little attention and most studies only consider multilevel modelling for its capacity to account for the clustering of patients within hospitals in order to obtain 'correct SEs' on regression coefficients and ORs. Some authors even conclude that hospital averages (ORs, observed/expected values) obtained from multilevel analyses gives similar results compared with traditional logistic regressions analyses. This situation is interpreted as an argument for keeping

the traditional logistic regression as the standard method for performing risk adjustment of hospital quality comparisons.[47–49] However, we do not agree with this opinion. The first reason is that the fixed effects approach does not explicitly inform on components of variance. The second reason is that the equivalence between traditional and multilevel regression results only occurs when the hospital GCE (ie, the clustering) is low and the number of patients at the hospitals is very high (ie, reliable estimation of hospital averages).[26] In other words, traditional non-multilevel analyses give similar results to the multilevel analysis only when the hospital differences are not relevant (ie, low VPC) and the patient load is very large in every hospital (which is rarely the case). In addition, hospital level variables appear paradoxically more statistically 'significant' when the hospital level is less relevant (ie, low VPC).[7] Information on the size of the hospital GCE is, therefore, fundamental for a sound analysis of hospital performance.

In this study we have applied the AUC to evaluate the hospital GCE. The AUC is a measure of discriminatory accuracy frequently used for gauging the performance of prognostic and screening markers in medicine[8 9] but it can also be used to quantify hospital GCE.[10 34] So far, many epidemiologists may not be familiar with the use of measures of components of variance like the VPC for binary outcomes.[30] However, the AUC measure is well established in clinical and healthcare epidemiology and the information it gives is relatively easy to interpret and communicate. From this perspective, the evaluation of hospital performance resembles a screening test and so we must therefore know the discriminatory accuracy of, for instance, a 'league table' to make informed decisions.

Analysing outcome indicators always requires adjustment for confounding (ie, patient-mix). We performed this adjustment using an innovative strategy that use hospitals and decile groups of RS in a cross-classified multilevel analysis. This approach provides a new option that could be very useful in some cases. In other cases, the classical inclusion of the patient-mix information as fixed effects may be more suitable. Both approaches provide similar results in terms of the hospital VPC, AUC and model fit. Then, the investigator may judge which approach is most suitable for the research question. A key advantage of our approach is it allowed a directed comparison of the importance of patient case mix and hospital effects in explaining variation in patient outcomes across hospital. We were able to calculate and compare a VPC for RS categories with the VPC for hospitals. We did not use existing patient-mix adjustment scores such as the Charlson Risk Score,[50] the Elixhauser Score[51] or the Center for Medicare & Medicaid Services Hierarchical Condition Category 'CMS-HCC' risk adjuster.[52] Our aim was not primarily to create a new RS equation but only to perform a parsimonious patient-mix adjustment where the RS summarises a large number of variables into a single construct.

This study has some limitations that need to be discussed. Unfortunately, the database for which we have ethical allowance for this study does not provide information on the severity of AMI or on revascularisation procedures (eg, percutaneous coronary intervention and coronary artery bypass grafting surgery). The inclusion of these variables could possibly improve the RS. However, we believe this improvement would be small and unlikely to affect our conclusions. Additionally, the RS is not a perfect instrument to quantify the true severity and mortality risk of a patient. Nevertheless, the RS categories we use are strongly associated with mortality and the RS alone shows a high discriminatory accuracy. RS may reflect practice or coding patterns of hospitals. However, Sweden has a very homogeneous healthcare system with centralised diagnostic rules which may reduce the risk of differential diagnosis setting. Finally, to explore the potential loss of information due to the categorisation of RS into deciles, we performed a sensitivity analysis with 15 and 20 categories. The results were similar to those obtained in model 2 (data not shown)

In summary, we illustrate the MAIHDA approach to auditing hospital performance using a three-step strategy. We argue that it is necessary to consider both the size of the hospital GCE and the RS adjusted 30-day mortality in relation to a pre-set benchmark value. Our results indicate that, at the time of our analysis, all hospitals in Sweden were performing homogeneously well. That is, there were no meaningful hospital differences and the benchmark target for 30-day mortality rate after admission for AMI was well achieved. Therefore, possible quality interventions should be universal and oriented to maintain the high hospital quality of care.

**Author affiliations**
[1]Unit for Social Epidemiology, Faculty of Medicine, Lund University, Malmö, Sweden
[2]Department of Public Health and Epidemiology, Pontificia Universidad Javeriana - Cali, Cali, Colombia
[3]Center for Primary Health Care Research, Region Skåne, Malmö, Sweden
[4]Institute of Health Management, Policy and Evaluation, University of Toronto, Toronto, Ontario, Canada
[5]Institute for Clinical Evaluative Sciences, Toronto, Ontario, Canada
[6]Schulich Heart Research Program, Sunnybrook Research Institute, Toronto, Ontario, Canada
[7]Centre for Multilevel Modelling, University of Bristol, Bristol, UK

**Contributors** JM had the initiative of the study and acquired the data. JM and MR-L wrote the original manuscript. MR-L and RP-V performed the analyses in coordination with JM. PA and GL provided advanced statistical support. All the authors have contributed to the design of the study, developed the methodology, direction of the analyses, interpretation of the results and drafting of the manuscript. All authors have revised the last version of the manuscript.

**Funding** The Swedish Research Council (Vetenskapsrådet) supported this work through the project 'Multilevel Analyses of Individual Heterogeneity: innovative concepts and methodological approaches in Public Health and Social Epidemiology': (#2017-01321, PI: Juan Merlo). Dr Austin received a personal Mid-Career Investigator award from the Heart and Stroke Foundation.

**Competing interests** None declared.

**Patient consent for publication** Not required.

**Provenance and peer review** Not commissioned; externally peer reviewed.

**Data availability statement** Data are available upon reasonable request. Original data are available from the Swedish National Board of Health and Welfare and Statistics, after approval of a research project by an Ethical Committee and by the

data safety committees of Swedish Authorities. However, it is possible to replicate the analyses performed, by using a table provided as supplementary information.

**ORCID iDs**
Merida Rodriguez-Lopez http://orcid.org/0000-0001-8245-0811
Juan Merlo http://orcid.org/0000-0001-8379-9708

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
