## [Reviewer comments · BMJ Open]

ARTICLE DETAILS

TITLE (PROVISIONAL)	A Cross-Classified Multilevel Analysis of Individual Heterogeneity and Discriminatory Accuracy (MAIHDA) to evaluate hospital performance: The case of Hospital Differences in Patient Survival after Acute Myocardial Infarction
AUTHORS	Rodriguez-Lopez, Merida; Merlo, Juan; Perez-Vicente, Raquel; Austin, Peter; Leckie, George

VERSION 1 – REVIEW

REVIEWER	Hayato Yamana The University of Tokyo, Japan
REVIEW RETURNED	03-Mar-2020

GENERAL COMMENTS	The authors applied a novel statistical approach and evaluated hospital performance in Sweden. The new method would be of interest to health services researchers evaluating institutional performance. However, as an Original Article, the manuscript would require more novelty in addition to a report that they tried a new method. There should be additional evaluations regarding either the methods itself or the applied results (hospital performance). Major comment: 1. The authors introduced the “cross-classified” MAIHDA, which treats both the risk score and hospitals as random effects. As far as I read from the introduction, this approach has not been applied in the evaluation of hospital performance. Therefore, advantages of the cross-classified approach over traditional multilevel approach could be one novelty of the work. However, it is important to confirm that the new method is better than, or at least comparable to, the “traditional” multilevel approach used commonly. I would suggest a comparison with a model that uses RS as a fixed effect and discussion based on the comparison. Minor comments: 2. RS was calculated using diagnosis records of diseases registered in the database. However, RS built this way may differ from the true severity of a patient in that recorded diagnoses may reflect practice or coding patterns of hospitals in the area. For example, a high-quality hospital may identify diabetes early whereas a patient with similar diabetes living in a different area may not be diagnosed. Therefore, hospital and RS may not be completely independent.3. Several variables may be added to improve the model, including other indices of patient severity, procedures (PCI, CABG, etc), and hospital characteristics. (However, as discussed in reference #5, an
---

	addition of hospital characteristics to a multilevel model with hospital-specific random effects may not increase the AUC). Because the hospitals seem to perform homogenously well according to the current model, there may not be a considerable change. However, authors should consider adding other variables, or discuss the limitation if other data were not obtainable. 4. Hospital and RS may have interactive effect. For example, patients with relatively mild conditions may have similarly good outcomes regardless of where they are treated, whereas outcomes of the most severe patients may be considerably affected by hospital performance. Such effect may be modeled in the “traditional” multilevel analysis using a random slope for RS. It would be interesting to know whether this effect can be modeled using the novel approach.
--	--

REVIEWER	Robert M West University of Leeds, UK
REVIEW RETURNED	13-Jul-2020

GENERAL COMMENTS	This is an interesting article with a good proposal for evaluating hospital (in general group) performance. The authors argue their use of random effects very well, highlighting advantages throughout. These advantages are justified. The authors answer the main objection that others might have concerning random effects. They do not however fully acknowledge the implied assumptions. In particular there is the assumption that the random effects are normally distributed. If they are then this approach is really efficient with all the advantages that the authors list. I certainly would like to make use of their methods. There could however be checks put in place to reassure the user that the normal assumption of each random effect is sufficiently satisfied. There are also further weaknesses that are revealed by exploring the method details. A single-level fixed effects model is used to create a risk score. Then there is discretisation into risk categories. Here there is potential to lose information. Then the categorised score is used as a random effect. This has the advantage of providing assessment in terms of VPCs and UACs. A simpler alternative would surely have been to fit a mixed effects regression with a random intercept for hospital. Why is this two-step approach needed? The hospital random effect then provides an assessment of hospital performance, and no information is lost due to categorisation. The authors should compare these approaches. Simple improvements to the work would be to include some sensitivity analyses by varying the number of risk categories, and providing R code as well as STATA code. It is not completely straightforward converting from STATA for a user who daily uses R and only occasionally STATA. The manuscript is easy to read and the logic is clearly expressed.
---

VERSION 1 – AUTHOR RESPONSE

Reviewer 1

The authors applied a novel statistical approach and evaluated hospital performance in Sweden. The new method would be of interest to health services researchers evaluating institutional performance.

However, as an Original Article, the manuscript would require more novelty in addition to a report that they tried a new method. There should be additional evaluations regarding either the methods itself or the applied results (hospital performance).

R/, Thank you for your positive opinion. We have now performed additional evaluations suggested by the reviewer below.

Major comment:

1. The authors introduced the “cross-classified” MAIHDA, which treats both the risk score and hospitals as random effects. As far as I read from the introduction, this approach has not been applied in the evaluation of hospital performance. Therefore, advantages of the cross-classified approach over traditional multilevel approach could be one novelty of the work. However, it is important to confirm that the new method is better than, or at least comparable to, the “traditional” multilevel approach used commonly. I would suggest a comparison with a model that uses RS as a fixed effect and discussion based on the comparison.

R/We thanks the reviewer’s comments. We have performed the traditional approach by including the RS categories in the fixed part in a Hierarchical Random Intercept Model (See the Stata output below). As expected, the results from both approaches are comparable in terms of Bayesian DIC, VPC and AUC (see table below).

Model characteristics	Random intercepts with patient RS as fixed effect	Hospital and risk score as random effect (cross classified)
Total AUC	0.78 (0.77- 0.79)	0.78 (0.77 – 0.79)
Hospital Variance	0.03(0.01-0.06)	0.03(0.01-0.06)
Bayesian DIC	2333.93	2333.54

The key advantage of the cross-classified (CC) multilevel approach over the traditional multilevel approach, is that it allows one to directly quantify the components of variance of both the hospital and the patient mix categories. Comparing the VPC of the patient mix (RS categories) with that of the hospital provides relevant information of the relative importance of the hospital general effect as compared with the effect of patient morbidity for understanding patient survival after AMI. This information also helps when evaluating the size of the general hospital effect (using the VPC).

We believe the CC approach provides a new option that could be very useful in some cases. In other cases, the classical inclusion of the patient-mix information as fixed effects (as the referee indicated) maybe more suitable.

The investigator may judge which approach is most suitable for the research question. We inform on those aspect in the discussion part of the revised manuscript.

Stata output (traditional)

```
. runmlwin prop cons risk10_2 risk10_3 risk10_4 risk10_5 risk10_6 risk10_7 risk10_8 risk10_9 risk
> 10_10, ///
> level3(hospital_new: cons, residuals(v, savechains("v2.dta", replace))) ///
> level2(stratum:) ///
> level1(stratum:) ///
> discrete(distribution(binomial) link(logit) denom(denom)) ///
> mcmc(cc burnin(5000) chain(10000) thin(10) hc(3) savechains("b2.dta", replace)) ///
> initsprevious nopause
```

```
MLwiN 3.05 multilevel model                        Number of obs      =       680
Binomial logit response model
Estimation algorithm: MCMC
```

Level Variable	No. of Groups	Observations per Group		
		Minimum	Average	Maximum
hospital_new	68	10	10.0	10
stratum	680	1	1.0	1

```
Burnin           =       5000
Chain            =      10000
Thinning        =         10
Run time (seconds) =       12.9
Deviance (dbar) =     2289.70
Deviance (thetabar) =    2245.46
Effective no. of pars (pd) =    44.23
Bayesian DIC    =    2333.93
```

proportion	Mean	Std. Dev.	ESS	P	[95% Cred. Interval]	
cons	-5.02	0.19	12	0.000	-5.42	-4.71
risk10_2	0.73	0.23	17	0.000	0.29	1.20
risk10_3	1.11	0.22	16	0.000	0.70	1.57
risk10_4	1.35	0.22	15	0.000	0.95	1.78
risk10_5	1.73	0.21	13	0.000	1.36	2.17
risk10_6	2.24	0.20	12	0.000	1.89	2.64
risk10_7	2.60	0.20	12	0.000	2.26	3.01
risk10_8	3.04	0.20	12	0.000	2.72	3.46
risk10_9	3.37	0.20	12	0.000	3.05	3.78
risk10_10	3.95	0.20	12	0.000	3.63	4.37

Random-effects Parameters	Mean	Std. Dev.	ESS	[95% Cred. Int]	
Level 3: hospital_new var(cons)	0.03	0.01	355	0.01	0.06

Obs	ROC Area	Std. Err.	-Asymptotic Normal- [95% Conf. Interval]	
43,247	0.7811	0.0038	0.77358	0.78854

Minor comments:

2)RS was calculated using diagnosis records of diseases registered in the database. However, RS built this way may differ from the true severity of a patient in that recorded diagnoses may reflect practice or coding patterns of hospitals in the area. For example, a high-quality hospital may identify

diabetes early whereas a patient with similar diabetes living in a different area may not be diagnosed. Therefore, hospital and RS may not be completely independent.

R/We agree with the reviewer that the RS is not a perfect instrument to quantify the true severity and mortality risk of a patient. We also agree that the ICD codes included in the equation may be influenced by the practice or coding patterns of hospitals. However, we note that Sweden has a very homogenous health care system with centralized diagnostic rules which may reduce the risk of differential diagnosis setting. Also, while the validity of certain ICD codes is high for some diseases it may be lower for other. A previous study provides detailed information on this subject

- Ludvigsson JF, Andersson E, Ekbom A, Feychting M, Kim JL, Reuterwall C, et al. External review and validation of the Swedish national inpatient register. *BMC Public Health*. 2011;11:450. doi: 10.1186/1471-2458-11-450. PubMed PMID: 21658213; PubMed Central PMCID: PMC3142234.

Nevertheless, the RS categories we use are strongly associated with mortality and the RS alone shows a high discriminatory accuracy. We have added this reflection in the discussion part of the revised manuscript.

3) Several variables may be added to improve the model, including other indices of patient severity, procedures (PCI, CABG, etc), and hospital characteristics. (However, as discussed in reference #5, an addition of hospital characteristics to a multilevel model with hospital-specific random effects may not increase the AUC). Because the hospitals seem to perform homogeneously well according to the current model, there may not be a considerable change. However, authors should consider adding other variables, or discuss the limitation if other data were not obtainable.

R/ The reviewer is right. Unfortunately, the database for which we have ethical allowance for this study does not include those variables (e.g., PCI, CABG). The inclusion of those variables would possibly improve the RS. However, we believe this improvement would be small and do not affect our conclusions.

Concerning hospital level variables - as the referee describes - they will not increase the discriminatory accuracy if the hospital level is included as a random effect in a multilevel regression. This is because the prediction used for calculating the AUC is the same but now the hospital effects are decomposed into fixed and a random effects components. By explaining the hospital level variance, the inclusion of hospital level variables helps us to understand the mechanism behind the hospital general effect.

We have now included this limitation at the end of the discussion section: "Unfortunately, the database for which we have ethical allowance for this study does not provide information on the severity of AMI or on revascularization procedures (e.g., PCI, CABG). The inclusion of these variables would possibly improve the RS. However, we believe this improvement would be small and unlikely to affect our conclusions. In any case, if the RS underestimates the true mortality risk of the patients, the already small hospital general effects would be viewed as even smaller".

4) Hospital and RS may have interactive effect. For example, patients with relatively mild conditions may have similarly good outcomes regardless of where they are treated, whereas outcomes of the most severe patients may be considerably affected by hospital performance. Such effect may be modeled in the "traditional" multilevel analysis using a random slope for RS. It would be interesting to know whether this effect can be modeled using the novel approach.

R/We thank the reviewer for this recommendation. We agree. Using a random slope in the traditional approach provides useful information on patient and hospital level heterogeneity. We have actually a publication using this approach

- Comendeiro-Maaloe M, Estupinan-Romero F, Thygesen LC, Mateus C, Merlo J, Bernal-Delgado E, et al. Acknowledging the role of patient heterogeneity in hospital outcome reporting: Mortality after acute myocardial infarction in five European countries. PLoS One. 2020;15(2):e0228425. Epub 2020/02/07. doi: 10.1371/journal.pone.0228425. PubMed PMID: 32027676; PubMed Central PMCID: PMC7004308.

We therefore agree that it is interesting to explore whether we can also allow for hospital by RS interactions using the CC approach. We have therefore fitted a third model, which additionally includes a random hospital by RS category interaction classification or in other words stratum random effect. This is an additional random effect which captures the deviation in AMI rates from those predicted by the main effects of hospital and RS. However, allowing for these random interaction effects adds little to the model. The stratum random effect variance is very small (less than a tenth of the size of the hospital random effect) and there were no improvements in the Bayesian DIC, suggesting that the hospital and RS random effects sufficiently capture the variability in the AMI rates. See the results below:

```
runmlwin prop cons, ///
  level4(hospital_new: cons, residuals(v)) ///
  level3(risk10: cons, residuals(u)) ///
  level2(stratum: cons, residuals(e)) ///
  level1(stratum:) ///
  discrete(distribution(binomial) link(logit) denominator(denom)) ///
  mcmc(cc burnin(1000) chain(10000) thin(1) hc(3)) ///
  initspreviousnopausecformat(%9.3f)

MLwiN 3.05 multilevel model          Number of obs   =   680
Binomial logit response model
Estimation algorithm: MCMC
```

Level Variable	No. of Groups	Observations per Group Minimum	Average	Maximum
hospital_new	68	10	10.0	10
risk10	10	68	68.0	68
stratum	680	1	1.0	1

```
Burnin           =   1000
Chain            =  10000
Thinning         =     1
Run time (seconds) =   14.8
Deviance (dbar)  =  2283.60
Deviance (thetabar) =  2231.98
Effective no. of pars (pd) =  51.61
Bayesian DIC     =  2335.21
```

proportion	Mean	Std. Dev.	ESS	P	[95% Cred. Interval]
cons	-3.007	0.446	9095	0.000	-3.910 -2.112

Random-effects Parameters	Mean	Std. Dev.	ESS	[95% Cred. Int]
Level 4: hospital_new var(cons)	0.035	0.013	590	0.015 0.065
Level 3: risk10 var(cons)	1.945	1.201	7769	0.729 5.027
Level 2: stratum var(cons)	0.003	0.004	11	0.000 0.019

We have included in the methods section this sentence: “While patients with relatively mild conditions may have similarly good outcomes regardless of where they are treated, outcomes of the most complex patients may be affected by hospital performance. We therefore fitted a cross-classified model including a random interaction effect between the Hospital and the RS (Supplementary material S2). That is, we allowed the effect that a Hospital has on their patients to vary according to the RS classification of their patients and vice versa. However, the resulting interaction classification variance was very low, suggesting that hospital attended and patient RS have additive effects on the log-odds of AMI. Consequently, we based our analysis on model 1 and 2.”

Reviewer:2

1) This is an interesting article with a good proposal for evaluating hospital (in general group) performance. The authors argue their use of random effects very well, highlighting advantages throughout. These advantages are justified. The authors answer the main objection that others might have concerning random effects. They do not however fully acknowledge the implied assumptions.

In particular, there is the assumption that the random effects are normally distributed. If they are then this approach is really efficient with all the advantages that the authors list. I certainly would like to make use of their methods. There could however be checks put in place to reassure the user that the normal assumption of each random effect is sufficiently satisfied

R/We appreciate the reviewer’s comments. The reviewer is right; in a multilevel model each set of predicted random effects should be approximately normally distributed and it is important to check this. In the figure below, we show the distribution of the predicted hospital and RS random effects for the cross-classified model. The plot for the predicted hospital random effects suggests the normality assumption is very reasonable, albeit with perhaps a suggestion of one slight outlying hospital. The plot for the predicted RS category random effects is always going to look somewhat noisy given that here we have lower number of random effects. The smoothed histogram is nonetheless reassuringly unimodal and given the small number of random effects it tracks the normal curve sufficiently well. We have now added the following sentence in the method section to communicate these points “Visual inspection of the hospital and RS category predicted random effects showed the random effect

normality assumptions weresatisfied". We have also included in the Stata dofile the commands that were used for generating the histograms shown below to encourage others to also assess the normality assumptions when applying our approach to their own data.

2) There are also further weaknesses that are revealed by exploring the method details. A single-level fixed effects model is used to create a risk score. Then there is discretization into risk categories. Here there is potential to lose information. Then, the categorised score is used as a random effect. This has the advantage of providing assessment in terms of VPCs and UACs. A simpler alternative would surely have been to fit a mixed effects regression with a random intercept for hospital. Why is this two-step approach needed? The hospital random effect then provides an assessment of hospital performance, and no information is lost due to categorization. The authors should compare these approaches.

R/ We believe a main question of interest when evaluating hospital performance is to quantify the general hospital effect on the individual risk of mortality after AMI after accounting for patient mix. For this purpose, we can use several strategies.

- 1.- The traditional approach is to perform a hierarchical/mixed effects/multilevel regression model including a random effect for the hospital and all identified patient's characteristics as fixed effects
- 2.- Alternatively we can use the strategy 1 but including a RS that summarizes all (or the fundamental) patient level variables

Strategies 1 and 2 both lead to an adjusted VPC_H that quantifies the General Hospital Effect. However, neither strategy provides the VPC associated with RS. This prompts our strategy.

3. The Cross-Classified Model where RS categories and hospital are both included as random effect. Only this strategy provides direct assessment in terms of VPCs and AUC, as the referee stress. We have compared the approaches and we provided a Stata do file for a better understanding. The conclusion of the different analytical approaches are – as expected – similar. Also see our response to Reviewer 1 on this point.

In relation to the RS categorization, we agree with the reviewer that the categorization of the risk score into deciles implies some loss of information. However, when including a continuous RS as a fixed effect we need to enter RS flexibly most likely as a polynomial. However, misspecification of this risk function will also produce some loss of information. So there are risks associated with traditional approach as well. The categorization of the RS, however, gives a very flexible alternative to the risk function as the risk associated with each category is freely estimated. When including the RS as a random effect, the RS needs to be categorized. In this case, the predicted RS random effect values

quantify the specific risks (analogous to the fixed effects). To gain information, we could increase the number of RS categories from, for example, 10 to 100, but this would also weaken the analyses as at least some “categories” will contain very few observations. We chose deciles as a compromise to provide enough granulation of the continuous RS variable and enough categories to be included as a random effect in the multilevel model.

3) Simple improvements to the work would be to include some sensitivity analyses by varying the number of risk categories

R/ As suggested by the reviewer we have performed a sensitivity analyses dividing the RS into 15 and 20 categories. The VPC and the AUC at hospital and RS levels were very similar between models. We have included this sentence at the end of the discussion section: “To explore the potential loss of information due to the categorization of RS into deciles, we performed a sensitivity analysis using 15 and 20 categories. The results were similar to those obtained in model 2 (data not shown).” The results are shown below:

15 categories of the RS

```
MLwiN 3.2 multilevel model           Number of obs   =   1020
Binomial logit response model
Estimation algorithm: MCMC
```

Level Variable	No. of Groups	Observations per Group		
		Minimum	Average	Maximum
hospital_new	68	15	15.0	15
risk15gr	15	68	68.0	68
stratum2	1020	1	1.0	1

```
Burnin           =   5000
Chain            =  10000
Thinning         =     1
Run time (seconds) =   22.9
Deviance (dbar)  =  3022.93
Deviance (thetabar) =  2973.52
Effective no. of pars (pd) =  49.41
Bayesian DIC     =  3072.34
```

proportion	Mean	Std. Dev.	ESS	P	[95% Cred. Interval]
cons	-2.997293	.3349914	8698	0.000	-3.652539 -2.318472

Random-effects Parameters	Mean	Std. Dev.	ESS	[95% Cred. Int]
Level 4: hospital_new var(cons)	.0351173	.01298	599	.0150966 .0653681
Level 3: risk15gr var(cons)	1.683541	.762337	7695	.7703128 3.586496

```

. roctab y Ph [fw=weight]
      Obs      ROC      Std. Err.      -Asymptotic Normal-
      Area      [95% Conf. Interval]
-----+-----
44,087,949    0.5711    0.0002    0.57083    0.57146

. roctab y Pr [fw=weight]
      Obs      ROC      Std. Err.      -Asymptotic Normal-
      Area      [95% Conf. Interval]
-----+-----
44,087,949    0.7782    0.0001    0.77793    0.77840

. roctab y Pt [fw=weight]
      Obs      ROC      Std. Err.      -Asymptotic Normal-
      Area      [95% Conf. Interval]
-----+-----
44,087,949    0.7856    0.0001    0.78541    0.78587

```

20 categories of the RS

```

MLwiN 3.2 multilevel model          Number of obs   =   1360
Binomial logit response model
Estimation algorithm: MCMC

```

Level Variable	No. of Groups	Observations per Group		
		Minimum	Average	Maximum
hospital_new	68	20	20.0	20
risk20gr	20	68	68.0	68
stratum3	1360	1	1.0	1

```

Burnin           =   5000
Chain            =  10000
Thinning         =     1
Run time (seconds) =  24.3
Deviance (dbar)  =  3615.16
Deviance (thetabar) = 3559.41
Effective no. of pars (pd) = 55.75
Bayesian DIC     =  3670.91

```

proportion	Mean	Std. Dev.	ESS	P	[95% Cred. Interval]
cons	-3.009225	.2886768	9147	0.000	-3.583359 -2.437632

Random-effects Parameters	Mean	Std. Dev.	ESS	[95% Cred. Int]
Level 4: hospital_new				
var(cons)	.0372363	.0136997	514	.0157387 .0686639
Level 3: risk20gr				
var(cons)	1.650638	.6241217	7740	.8355834 3.196447

```

. roctab y Ph [fw=weight]

      Obs      ROC      Std. Err.      -Asymptotic Normal—
      Area      [95% Conf. Interval]
-----+-----
58,791,516  0.5693  0.0001  0.56900  0.56954

. roctab y Pr [fw=weight]

      Obs      ROC      Std. Err.      -Asymptotic Normal—
      Area      [95% Conf. Interval]
-----+-----
58,791,516  0.7795  0.0001  0.77926  0.77967

. roctab y Pt [fw=weight]

      Obs      ROC      Std. Err.      -Asymptotic Normal—
      Area      [95% Conf. Interval]
-----+-----
58,791,516  0.7866  0.0001  0.78644  0.78684

```

4)..and providing R code as well as STATA code. It is not completely straightforward converting from STATA for a user who daily uses R and only occasionally STATA. The manuscript is easy to read and the logic is clearly expressed.

R/ We have now attached an R script as supplemental material (S3).

VERSION 2 – REVIEW

REVIEWER	Hayato Yamana The University of Tokyo, Japan
REVIEW RETURNED	13-Aug-2020

GENERAL COMMENTS	The concerns that I had raised were properly addressed through additional analyses and revisions. Regarding Major comment 1, please add the methods/results of the traditional approach (RS categories as the fixed part) into the manuscript. The discussion on p.20-21 should follow the results.
--

REVIEWER	Robert M West University of Leeds, UK
REVIEW RETURNED	12-Aug-2020

GENERAL COMMENTS	The authors have addressed the issues raised by the reviewers. Specifically the Gaussian distribution of random effects has been acknowledged and the categorization of the risk score has been explored with sensitivity analysis.
---